# Glycerol-3-phosphate mediates rhizobia-induced systemic signaling in soybean

M.B. Shine[1,3], Qing-ming Gao[1,3], R.V. Chowda-Reddy[2], Asheesh K. Singh [2], Pradeep Kachroo [1] & Aardra Kachroo[1]*

Glycerol-3-phosphate (G3P) is a well-known mobile regulator of systemic acquired resistance (SAR), which provides broad spectrum systemic immunity in response to localized foliar pathogenic infections. We show that G3P-derived foliar immunity is also activated in response to genetically-regulated incompatible interactions with nitrogen-fixing bacteria. Using gene knock-down we show that G3P is essential for strain-specific exclusion of non-desirable root-nodulating bacteria and the associated foliar pathogen immunity in soybean. Grafting studies show that while recognition of rhizobium incompatibility is root driven, bacterial exclusion requires G3P biosynthesis in the shoot. Biochemical analyses support shoot-to-root transport of G3P during incompatible rhizobia interaction. We describe a root-shoot-root signaling mechanism which simultaneously enables the plant to exclude non-desirable nitrogen-fixing rhizobia in the root and pathogenic microbes in the shoot.

---

[1] Department of Plant Pathology, University of Kentucky, Lexington, KY 40546, USA. [2] Department of Agronomy, Iowa State University, Ames, IA 50011, USA. [3]These authors contributed equally: MB Shine, Qing-ming Gao *email: apkach2@uky.edu

                                    1

Legume plants form symbiotic relationships with diazotrophic bacteria called rhizobia[1]. During such symbiosis, plants provide bacteria with preferred carbon sources such as malate and succinate in return for essential reduced nitrogen[2]. Successful symbiosis results in root nodules, which are specialized plant organs containing the optimal environment for rhizobia to convert nitrogen into ammonia. Bacterial entry into the host cells is dependent upon evasion of the first tier of host immunity and recognition as beneficial by the host[3–6]. Strategies for evading host immunity include suppression of host defenses via bacterial surface lipopolysaccharides and secreted proteins[5–7]. Some rhizobia also inject effectors into the host cell via the type III secretion system, a well-known virulence strategy used by pathogenic bacteria[8]. Compatibility occurs when legume root-exuded flavonoids induce the production of Nod factors (lipochitooligosaccharides) in compatible rhizobia, which are in turn recognized by membrane-localized LysM-type receptor kinases called nodulation (Nod) factor receptors in the host[9]. In addition to Nod factors, bacterial exopolysaccharides are also important for the infection process and in some cases recognition of specific exopolysaccharide structures by host LysM receptor kinase can determine compatibility[10]. Compatible interactions result in a series of plant root modifications that eventually result in nodule formation. Bacteria living in the nodule cells fix nitrogen via the nitrogenase enzyme complex. The host tightly regulates the density of nodules by compatible bacteria based on nitrogen availability and a systemic signaling mechanism called autoregulation of nodulation (AON)[11]. AON involves recognition of root-synthesized CLE (CLAVATA3/endosperm-surrounding region) peptides by shoot-derived receptor complexes comprising a leucine-rich repeat containing receptor kinase, and the eventual inhibition of nodulation by a shoot-derived signal. Both miR2111 and cytokinin have been implicated as the potential shoot-derived signal[12,13].

Interestingly, as in plant-pathogen interactions, incompatibility in legume-rhizobia associations is also regulated in a genotype-specific manner[14]. The process likely involves recognition of bacterial effectors by host receptor proteins similar to the perception of pathogenic microbes during the second tier of plant immunity termed effector-triggered immunity (ETI)[15]. For example, genetic variations in the rhizobium NopP effector were recently shown to contribute to Rj2-mediated exclusion of incompatible rhizobia in soybean[16], suggesting that recognition of NopP by Rj2 determines incompatibility. Rj2 along with Rj3, Rj4 and Rfg1, was identified as a naturally occurring variant in soybean that restricts nodulation by specific bacterial strains[17–21]. The dominant Rj/Rfg genes, which are structurally different from the recessive Nod-factor receptors[22,23], are presumed to help exclude poor nitrogen-fixing or less-beneficial rhizobia such as the exclusion of B. japonicum USDA122 (U122) by Rj2 and Sinorhizobium fredii USDA257 (U257) or USDA205 by Rfg1[19,21]. Knockout mutations in Rj2 or Rfg1 promote nodule formation by the respective incompatible bacterial strains[24]. Rj2 and Rfg1 encoded proteins show homology to Toll interleukin-like receptor nucleotide-binding site leucine-rich repeat (TIR-NB-LRR) type of resistance (R) proteins, which are well-known regulators of ETI[25]. Interestingly, Rj2-mediated recognition of the NopP effector is associated with induction of the pathogenesis-related (PR) 2 gene[16], suggesting involvement of ETI like responses.

ETI, which provides race-specific resistance to pathogens at the site of pathogen entry, is induced when strain-specific avirulent (Avr) proteins from the pathogen associate directly/indirectly with cognate plant R proteins[15]. ETI can also result in the induction of a systemic immune response termed systemic acquired resistance (SAR), which protects the plant against secondary infections by related/unrelated pathogens at the whole plant level[26,27]. SAR signaling is dependent on the generation of mobile signals at the primary infection site, which translocate to distal tissue and prepare the plant against future pathogen infections. In Arabidopsis, SAR signaling is dependent on a number of chemical signals including salicylic acid (SA)[28], azelaic acid (AzA)[29], G3P[30], the free radicals nitric oxide (NO) and reactive oxygen species (ROS)[31,32], and pipecolic acid (Pip)[33]. The signaling mediated by these chemicals is organized into two parallel branches with G3P functioning downstream of Pip, NO/ROS, and AzA in one of these branches[26,27,34] (Supplementary Fig. 1). We show that genetic exclusion of incompatible rhizobia in the root requires conserved molecular components of ETI and results in the induction of systemic signaling, which involves SAR-associated chemical signals. Root recognition of incompatible rhizobia involves the generation of an unknown root signal, which travels to the shoot to induce the accumulation of the SAR-inducer G3P, thereby activating foliar resistance to pathogens. Transport of G3P back to the root enables root exclusion of rhizobia.

## Results

**Rhizobia incompatibility induces systemic pathogen resistance.** Based on the structural similarity of Rj2/Rfg1 to R proteins and the potential activation of ETI like responses during Rj2-derived signaling, we tested whether Rj2/Rfg1 functioned like ETI related R proteins and recruited molecular components typically involved in the R-mediated immune response[35–37]. Using previously generated VIGS (virus-induced gene silencing) vectors[38–41] we knocked down the expression of RAR1 (required for Mla12-mediated resistance), SGT1 (suppressor of the G2 allele of skp1), Hsp90 (heat shock protein 90), NDR1 (non-disease resistance 1), EDS1 (enhanced disease susceptibility 1), and PAD4 (phytoalexin deficient 4) in L76-1988 (Rj2 rfg1) and L82-2024 (rj2 Rfg1) soybean plants (Supplementary Fig. 2) followed by the evaluation of nodulation in response to compatible and incompatible rhizobia strains. Plants infected with empty VIGS vector[38] (V) were used as a control. As expected, root hair of U122-inoculated plants exhibited bending and curling in comparison to root hair of plants inoculated with buffer alone (Supplementary Fig. 3a). U122-induced root hair curling, cortical cell division and active nodule formation (based on pink coloration associated with the presence of leghemoglobin[42]) in rj2 Rfg1 plants (Fig. 1a, b, Supplementary Fig. 3b). U122 also induced root hair curling in Rj2 rfg1 plants but produced fewer nodule primordia that did not develop completely and did not produce nodules on Rj2 rfg1 plants (Fig. 1a–c, Supplementary Fig. 3b). There were no obvious differences in the number or size of nodules produced by U122 or U257 in any of the knockdown plants in their respective compatible backgrounds (Supplementary Fig. 4a). Interestingly, however, incompatible strains induced root hair curling and nodule primordia, eventually resulting in active nodules in plants knocked down for the defense-related components, RAR1 and NDR1 (Fig. 1, Supplementary Figs. 3 and 4). The U122-inoculated Rj2 rfg1 RAR1-(Sil_{RAR1}) and Rj2 rfg1 NDR1- (Sil_{NDR1}) knockdown plants showed comparable root hair curling as rj2 Rfg1 and Rj2 rfg1 V plants. These plants also showed comparable nodule primordia per cm root, as rj2 Rfg1 plants (Fig. 1c). This was consistent with the rj2 Rfg1 V-like number, density (per root) and size of nodules on Rj2 rfg1 Sil_{RAR1} and Rj2 rfg1 Sil_{NDR1} plants (Fig. 1, Supplementary Figs. 3 and 4). Notably, the requirement for NDR1, is an exception for TIR-NB-LRR type of R proteins, which typically recruit EDS1, rather than NDR1[36]. Although no strong conclusions can be made about the stage at which lack of RAR1/NDR1 interferes with Rj2/Rfg1-derived incompatibility, these results suggest that Rj2/Rfg1-derived exclusion of incompatible rhizobia requires molecular

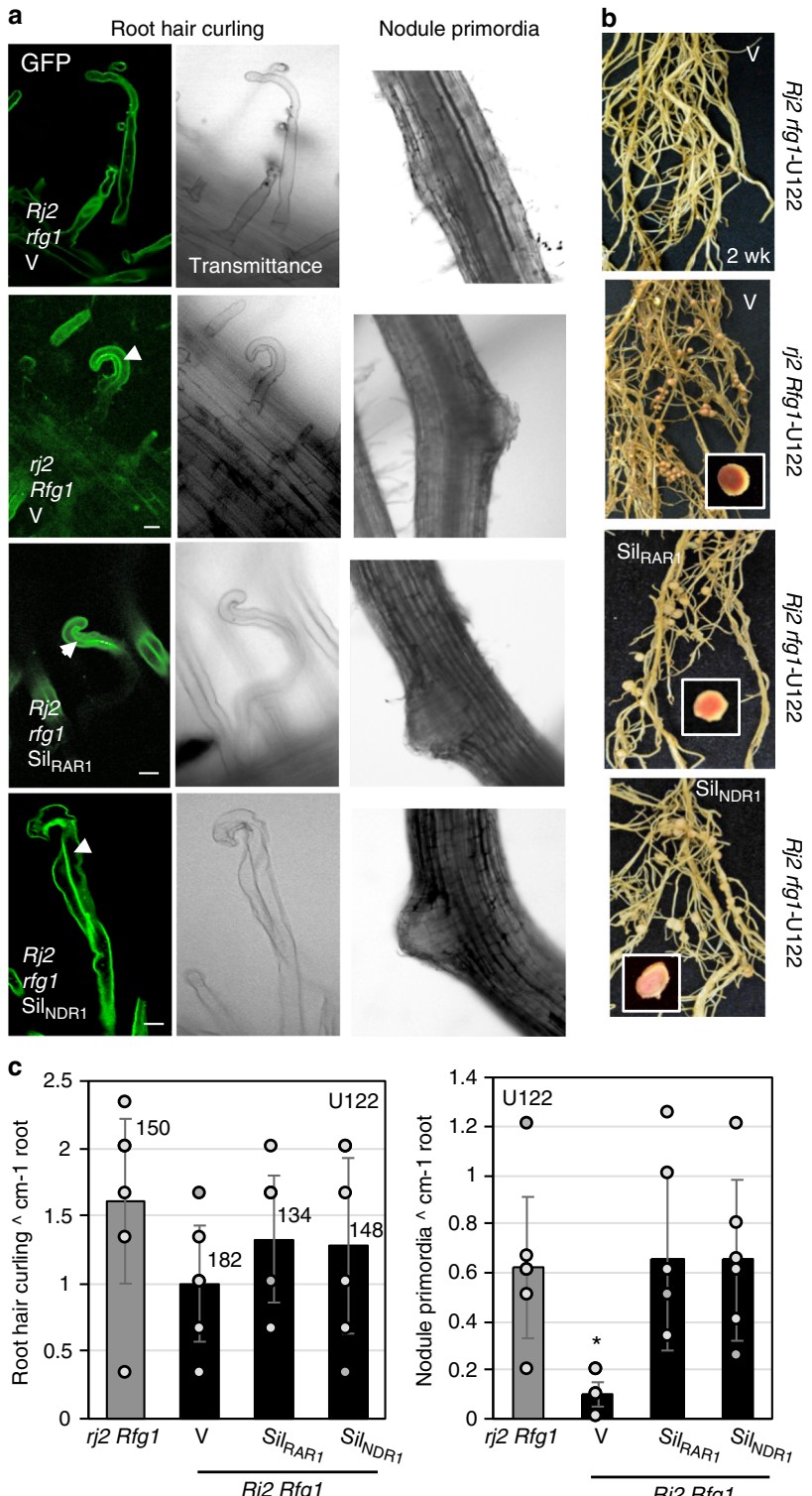

**Fig. 1** *Rj2*-derived rhizobia incompatibility involves *R*-mediated signaling components. **a** Micrographs (×40 magnification) of STYO®13 stained roots from *rj2 Rfg1* or *Rj2 rfg1* plants that were infected with control VIGS vector (V), or knocked down for GmNDR1 (Sil$_{NDR1}$) or GmRAR1 (Sil$_{RAR1}$) at 3 (root hair curling) or 6 (nodule primordia) days post inoculation (dpi) with U122. Images observed under GFP and transmittance channels are shown. Arrowhead indicates infection thread. Scale bars represent 270 microns. **b** Morphological phenotype of nodules produced by U122 or U257 on V, Sil$_{RAR1}$ and Sil$_{NDR1}$ plants of indicated genotype at 2 weeks post inoculation. Insets show cut nodules with pink coloration indicative of active nitrogen fixation. **c** Average number of curled root hairs (2–3 dpi) and nodule primordia (6 dpi) per cm root of U122-inoculated *rj2 Rfg1*, *Rj2 Rfg1* V (VIGS control), *Rj2 rfg1* Sil$_{RAR1}$ (RAR1-knockdown), and *Rj2 rfg1* Sil$_{NDR1}$ (NDR1-knockdown) plants. Differences in root hair curling for *rj2Rfg1* and *Rj2rfg1*V are not statistically significant at $P = 0.05$. Numbers above bars indicate the average of total number of root hairs per cm. For nodule primordia, asterisk denotes data significantly different from *rj2 Rfg1*, Student's *t*-test, $P < 0.05$. Results are representative of two independent experiments.

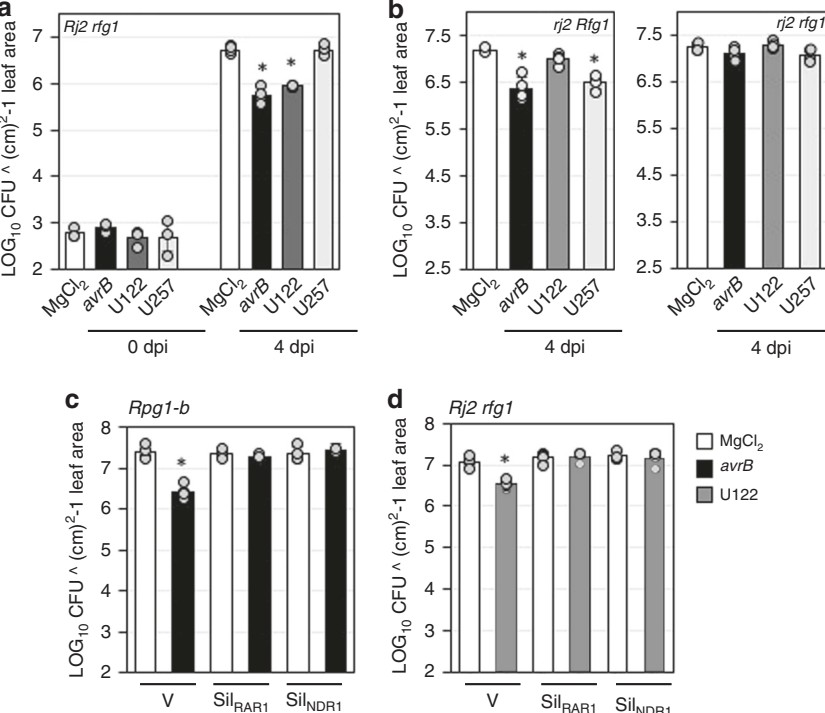

**Fig. 2** *Rj2*-derived rhizobia incompatibility induces systemic resistance in soybean. **a–d** Bacterial counts of *Psg Vir* in soybean plants that were pre-infiltrated (leaf) with buffer (MgCl$_2$) or *Psg avrB* (*avrB*), or pre-inoculated (root) with U122 or U257. LOG$_{10}$ values of colony forming units (CFU) ˆ cm$^2$-1 leaf area from infected leaves at 0 and 4 days post inoculation (dpi) are presented. Error bars indicate standard deviation ($n = 4$). Asterisks denote significant difference from MgCl$_2$ in (**a**) and (**b**), and significant difference from MgCl$_2$ of same genotype in (**c**) and (**d**), Student's *t*-test, $P < 0.0002$. **a** Wild-type plants of *Rj2 rfg1* genotype were used. **b** Wild-type plants of *rj2 Rfg1* or *rj2 rfg1* genotype were used. **c** V (VIGS vector control), S$_{RAR1}$ (*GmRAR1* knockdown) or S$_{NDR1}$ (*GmNDR1* knockdown) plants of *Rpg1-b* genotype were used. **d** V, S$_{RAR1}$ or S$_{NDR1}$ plants of *Rj2 rfg1* genotype were used. Results are representative of three-four independent experiments.

components that typically function in R-mediated defense against microbes.

R-mediated resistance is often associated with the induction of SAR. Therefore, we next assayed foliar resistance in response to Rj2/Rfg1-mediated control of nodulation. Interestingly, root inoculation with incompatible rhizobia resulted in the induction of foliar resistance to *Pseudomonas syringae* pv. *glycinea* (*Psg*); *Rj2 rfg1* plants root-inoculated with the incompatible U122 exhibited enhanced resistance to virulent *Psg* (*Psg Vir*, Fig. 2a). Likewise, *rj2 Rfg1* plants exhibited enhanced resistance to *Psg Vir* when pre-inoculated with the incompatible U257 (Fig. 2b). Neither U122 nor U257 induced foliar resistance in the respective compatible backgrounds (*rj2 Rfg1* and *Rj2 rfg1*, respectively) or in *rfg1 rj2* plants that were compatible to both rhizobia strains (Fig. 2b, right panel). Notably, incompatible rhizobia-induced foliar resistance was as robust as pathogen-induced SAR; systemic resistance induced upon pre-exposure to the avirulent pathogen *Psg avrB*. Incompatible rhizobia-induced foliar resistance within 24 h and this effect lasted at least 72 h post root inoculation (Supplementary Fig. 5). We next examined if systemic resistance induced in response to incompatible rhizobia was associated with Rj2 function. For this, we tested whether U122 could induce systemic foliar resistance in Sil$_{RAR1}$ and Sil$_{NDR1}$ plants because these components are known to directly contribute to R protein function[43]. Consistent with the demonstrated involvement of RAR1 and NDR1 in pathogen-induced SAR[39,44] as well as their requirement in Rpg1-b (specifies resistance to *Psg avrB*) function[39,40], both Sil$_{RAR1}$ and Sil$_{NDR1}$ *Rpg1-b* plants were unable to induce SAR in response to *Psg avrB* (Fig. 2c). Notably, Sil$_{RAR1}$ and Sil$_{NDR1}$ *Rj2* plants also failed to induce foliar pathogen resistance in response to the incompatible U122 (Fig. 2d).

Together, these results substantiated the notion that incompatible rhizobia-induced systemic resistance was associated with Rj2 function which required conserved signaling components of the ETI pathway.

**SAR inducers accumulate during rhizobia incompatibility.** Next, we examined whether the systemic resistance induced by incompatible rhizobia involved similar signaling mechanisms as pathogen-induced SAR. Functional conservation of SAR signaling between Arabidopsis and soybean was evident from the result that petiole exudate (Pex) collected from leaves of *Psg avrB*-infected (Pex$_{avrB}$), but not buffer-inoculated (Pex$_{MgCl_2}$) soybean plants was able to confer robust systemic resistance on *Arabidopsis thaliana* (Fig. 3a). Arabidopsis (Col-0) plants were leaf-infiltrated with MgCl$_2$ (control), *P. syringae* pv. *tomato* (*Pst*) *avrRpt2*, or soybean Pex$_{avrB/MgCl_2}$ followed by infection with *Pst* DC3000 on systemic leaves. Growth of DC3000 was significantly lower in plants pre-infiltrated with *Pst avrRpt2* or Pex$_{avrB}$ as compared with MgCl$_2$ or Pex$_{MgCl_2}$ (Fig. 3a). Measurement of SAR-associated metabolites showed that leaves of soybean plants accumulated G3P, AzA, and SA after localized infection with *Psg avrB* (Fig. 3b–d). Interestingly, root inoculation with incompatible (U122) rhizobia also increased G3P, AzA, SA, and ROS containing hydroxyl and carbon-centered radicals in the leaves of soybean plants (Fig. 3e–h). By comparison, root infection with the compatible strain U257 led to a nominal or no increase in G3P, AzA, SA or ROS levels in the leaves (Fig. 2e–h). Consistent with these results, Pex from leaves of U122 (Pex$_{U122}$), but not U257 (Pex$_{U257}$) inoculated *Rj2 rfg1* soybean plants, was able to induce robust systemic resistance in Arabidopsis plants (Fig. 3i). Together, these results suggested that incompatible rhizobia triggered systemic

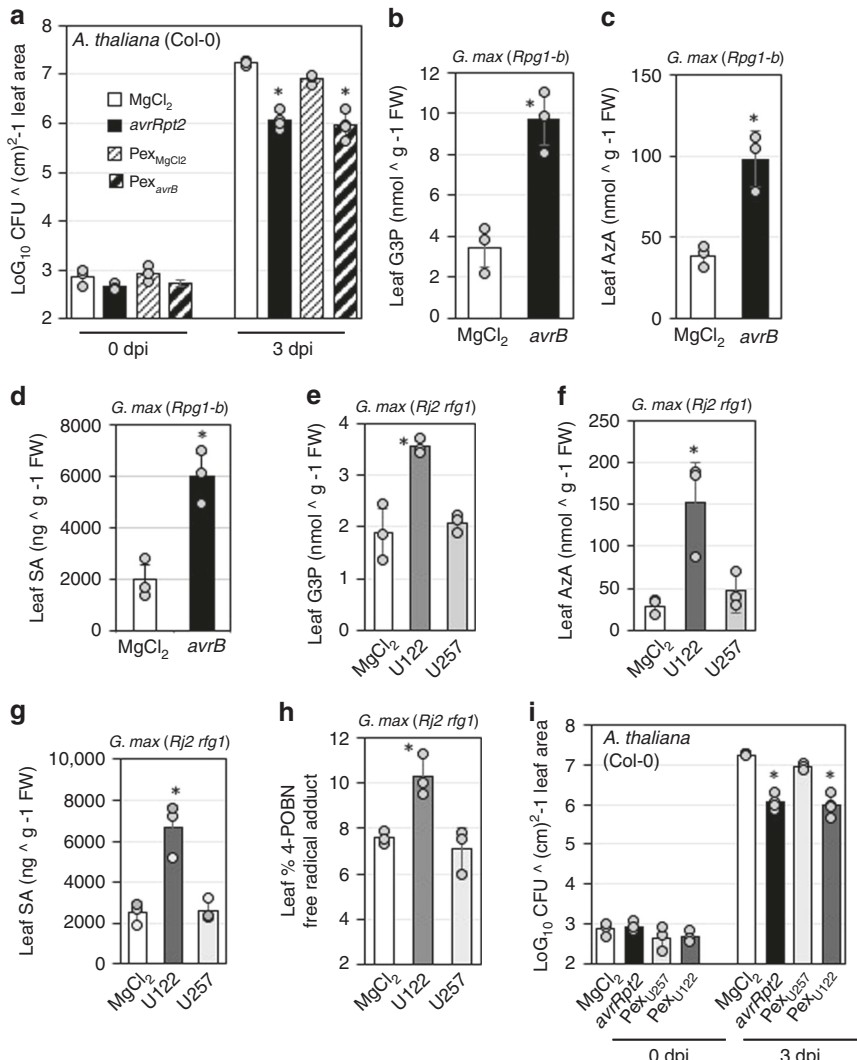

**Fig. 3** Incompatible rhizobia induce the accumulation of SAR-associated metabolites. **a** Bacterial counts of *Pst* DC3000 in *Arabidopsis thaliana* (Col-0). LOG$_{10}$ values of colony forming units (CFU) ˆ cm$^2$-1 leaf area from infected leaves at 0 and 4 dpi are presented. Error bars indicate standard deviation ($n =$ 4). Asterisks denote data significantly different from MgCl$_2$, Student's *t*-test, $P < 0.001$. Plants were pretreated with buffer (MgCl$_2$), *Pst avrRpt2*, or petiole exudate from MgCl$_2$-infiltrated (Pex$_{MgCl2}$) or *Psg avrB*-infected (Pex$_{avrB}$) soybean plants. **b–d** SAR-associated metabolite levels in MgCl$_2$ or *avrB* infected leaves of *Rpg1-b* soybean (*G. max*) plants at 24 h. **b** Glycerol-3-phosphate (G3P), **c** azelaic acid (AzA), and **d** salicylic acid (SA) levels. Error bars indicate standard deviation ($n = 5$). Asterisks denote data significantly different from MgCl$_2$, Student's *t*-test, $P < 0.0001$. **e–h** SAR-associated metabolite levels in leaves of MgCl$_2$, U122 or U257 inoculated *Rj2 rfg1* soybean (*G. max*) plants. **e** G3P at 24 h, **f** AzA at 24 h, **g** total SA at 48 h, **h** electron spin resonance (ESR) spectrometry showing relative levels of free radicals at 24 h post rhizobia inoculation. Error bars indicate SD ($n = 5$). Asterisks denote significant differences from MgCl$_2$, Student's *t*-test, $P < 0.005$. **i** Bacterial counts of *Pst* DC3000 in *A. thaliana* (Col-0). LOG$_{10}$ values of CFU ˆ cm$^2$-1 leaf area from infected leaves at 0 and 3 dpi are presented. Error bars indicate standard deviation ($n = 4$). Asterisks denote data significantly different from MgCl$_2$, Student's *t*-test, $P < 0.001$. Plants were pretreated with MgCl$_2$, *Pst avrRpt2*, or petiole exudate from U122$_2$ (Pex$_{U122}$) or U257 (Pex$_{U257}$) inoculated soybean plants. Results are representative of three independent experiments.

signaling, which overlapped with SAR in terms of increased accumulation of SAR-associated signals in the distal tissue.

**Rhizobia incompatibility induces distal transcriptional changes**. To better understand the mechanism underlying incompatible rhizobia-induced systemic resistance, we compared transcriptional changes in the foliar tissue of *Rj2* plants inoculated with buffer (MgCl$_2$), U122 or U257, using RNA-Seq analysis (Supplementary Fig. 6a). Interestingly, despite the significant overlap in their transcriptional changes (Supplementary Fig. 6a, Supplementary Data 1 and 2), plants inoculated with compatible versus incompatible rhizobia showed completely opposite nodulation and systemic resistance phenotypes (Supplementary

Fig. 4a, Fig. 2a). Importantly, despite significant overlap in metabolite accumulation, the transcriptional changes in response to incompatible rhizobia (Supplementary Data 3 and 4) did not significantly overlap with those observed in response to *Psg* infection[45]. Furthermore, genes associated with photosynthesis, photorespiration and primary metabolism are downregulated, and biotic stress and signaling related genes are induced in the systemic tissues of SAR-activated plants[46]. These trends were not observed in the foliar tissue of incompatible rhizobia-inoculated plants (Supplementary Data 3–6). Nonetheless, consistent with induction of SAR-associated chemicals, the transcriptional profiles induced in response to incompatible rhizobia did show some overlap with pathogen-induced responses such as induction of defense-related genes including pathogenesis-related 1 (*PR1*, SA

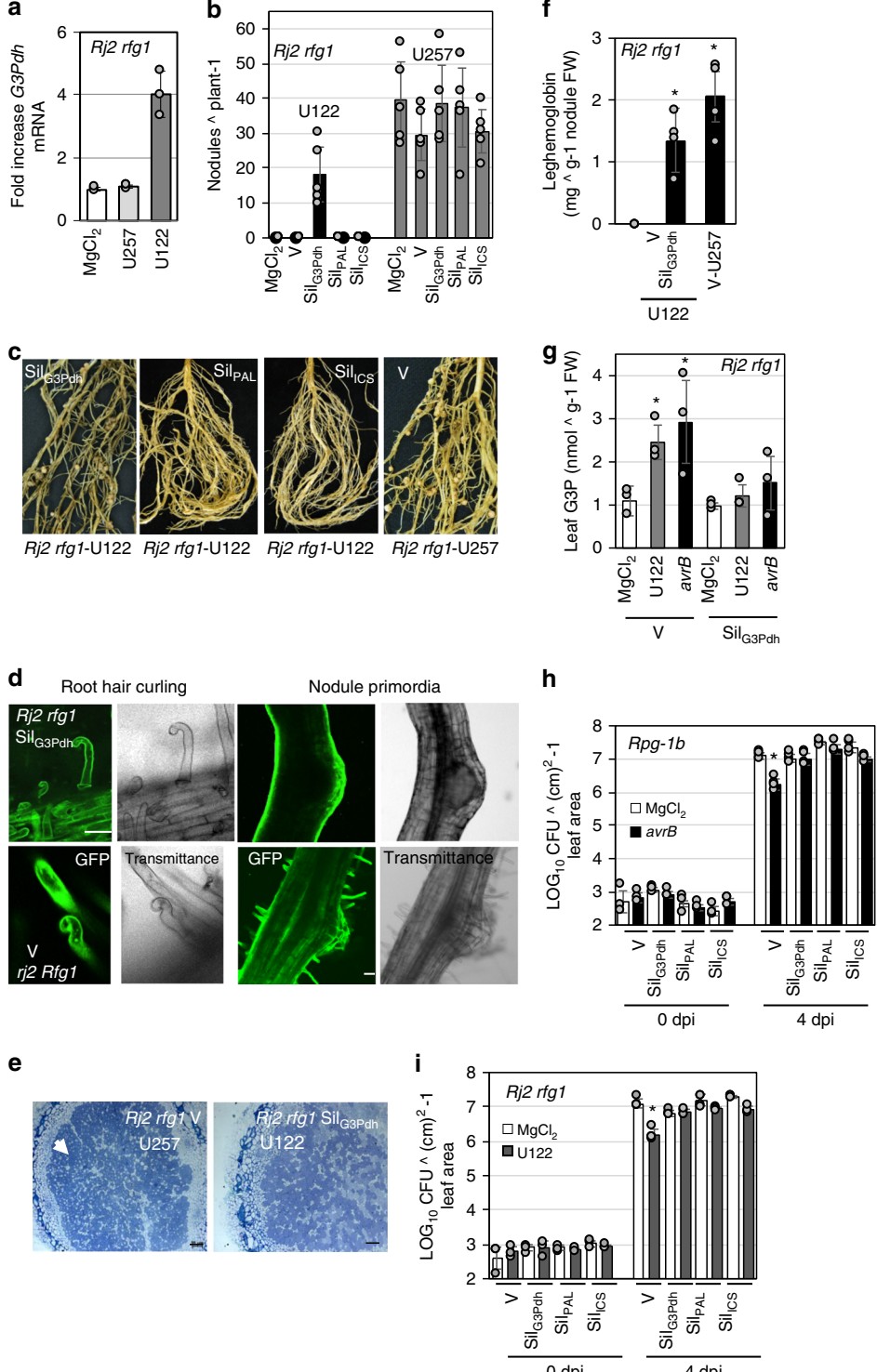

marker), nitrate reductase (*NR*, NO biosynthesis), and several ROS detoxifying enzymes (data shown for dihydroflavanol reductase, *DFR*) (Supplementary Fig. 6b). Notably, U122 inoculation also induced a putative G3P dehydrogenase (G3Pdh) *Glyma02g38310* in leaf tissue (Fig. 4a), implicating a role for G3P in U122-mediated foliar resistance and/or root nodulation. *Glyma02g38310* transcripts were expressed in both foliar and root tissues of soybean plants (Supplementary Fig. 7a) and *Glyma02g38310* complemented the growth defects of the yeast G3Pdh mutant *gpd1* (Supplementary Fig. 7b). This indicated that Glyma02g38310 is a functional G3Pdh.

**Rhizobia incompatibility requires glycerol-3-phosphate**. The above results prompted us to investigate the role of G3P in nodulation. We also examined whether SA contributed to *Rj2* incompatibility against U122 because G3P and SA function in two parallel branches of the SAR pathway (Supplementary Fig. 1[32]). SA biosynthesis is regulated by ICS (isochorismate synthase) and PAL (phenylalanine ammonia lyase) enzymes, both of which play an equally important role in SA biosynthesis in soybean[47]. Knockdown of *ICS* or *PAL* expression (Supplementary Fig. 2) did not alter *Rj2*-mediated incompatibility to U122 or compatibility to U257 (Fig. 4b, c). Interestingly, however, G3Pdh

**Fig. 4** Glycerol-3-phosphate is essential for incompatible rhizobia-induced systemic resistance. **a** Fold increase in mRNA levels of *Glyma02g38310* in the leaves of buffer (MgCl₂), U257 or U122-inoculated *Rj2 rfg1* soybean plants. Error bars indicate SD ($n = 3$). Asterisks denote data significantly different from MgCl₂, Student's *t*-test, $P < 0.0001$. **b** Average number of nodules produced by U122 or U257 on *Rj2 rfg1* plants inoculated with MgCl₂, control VIGS vector (V), or knocked down for G3Pdh (Sil$_{G3Pdh}$), PAL (Sil$_{PAL}$) and ICS (Sil$_{ICS}$). Error bars indicate SD ($n = 15$). Asterisks denote significant differences from mock, Student's *t*-test, $P < 0.0001$. Black circles on *x*-axis indicates absence of nodules. **c** Root morphology and nodule phenotypes of U122-inoculated *Rj2 rfg1* Sil$_{G3Pdh}$, Sil$_{PAL}$ or Sil$_{ICS}$ plants at 2 weeks post inoculation. **d** Root hair curling, infection thread (white arrow) and nodule primordia formation in V or Sil$_{G3Pdh}$ plants (*rj2 Rfg1* or *Rj2 rfg1*, respectively) at 2 (root hair curling) or 6 (nodule primordia) days post inoculation (dpi) with U122. Micrographs (×40 magnification) of STYO®13 stained roots observed under GFP and transmittance channels are shown. Scale bars represent 270 microns. **e** Micrographs (×40) of trypan blue stained nodules isolated from V or Sil$_{G3Pdh}$ plants (*Rj2 rfg1*) infected with U257 or U122, respectively, showing presence of live bacteria (white arrowhead). Scale bars represent 30 microns. **f** Leghemoglobin levels in V or Sil$_{G3Pdh}$ plants (*Rj2 rfg1*) inoculated with U122 or U257. Error bars indicate SD ($n = 3$). Asterisks denote significant differences from U122-inoculated V, *t*-test, $P < 0.0001$. **g** G3P levels in leaves of V and Sil$_{G3Pdh}$ plants (*Rj2 rfg1*) after leaf infiltration with MgCl₂ or *Psg avrB* or root inoculation with U122. Asterisks denote data significantly different from MgCl₂, Student's *t*-test, $P < 0.0001$. **h, i** Bacterial counts of *Psg Vir* in V, Sil$_{G3Pdh}$, Sil$_{PAL}$ or Sil$_{ICS}$ plants (*Rpg1-b* or *Rj2 rfg1*). LOG$_{10}$ values of colony forming units (CFU) ˆ cm²⁻¹ leaf area from *Psg Vir* infected leaves at 0 and 4 days post inoculation (dpi) are presented. Error bars indicate standard deviation ($n = 4$). Asterisks denote data significantly different from MgCl₂-treated plants of corresponding genotype, Student's *t*-test, $P < 0.001$. **h** Plants were leaf-infiltrated with MgCl₂ or *Psg avrB* followed by *Psg Vir* inoculation on systemic leaves. **i** Plants were root-inoculated with MgCl₂ or incompatible U122 followed by *Psg Vir* inoculation on leaves. Results are representative of two-four independent experiments.

knockdown (Sil$_{G3Pdh}$) *Rj2* plants (Supplementary Fig. 8a) did produce nodules in response to the incompatible U122 (Fig. 4b, c). The Sil$_{G3Pdh}$ plants expressed wild-type-like levels of *Rj2* or *Rfg1* in their roots (Supplementary Fig. 8b), suggesting that the breakdown of rhizobium incompatibility was not due to altered expression of *R* genes. Microscopic analysis of the infection process showed that U122-induced root hair deformations, infection thread formation, and cortical cell division in *Rj2 rfg1* Sil$_{G3Pdh}$ plants similar to that in *rj2 Rfg1* V plants, indicative of a fully compatible interaction (Fig. 4d). The U122-infected *Rj2 rfg1* Sil$_{G3Pdh}$ plants were comparable to *rj2 Rfg1* V plants in terms of the extent of root hair curling and nodule primordia per cm root as well as their nodule density per root, nodule size and the average number of nodules per plant (Fig. 4b, Supplementary Fig. 9). Furthermore, viable bacteria were detected in the fully formed nodules on U122-infected *Rj2 rfg1* Sil$_{G3Pdh}$ roots, similar to U257-infected *Rj2 rfg1* V roots (Fig. 4e). The Sil$_{G3Pdh}$ nodules expressed transcripts for the bacterial dinitrogenase reductase gene *nifH1*[48] (Supplementary Fig. 10a), and accumulated the hemoprotein leghemoglobin (regulates oxygen levels to promote nitrogenase activity and bacterial growth in the nodules) (Fig. 4f). This correlated with the increased expression of nodulation-specific genes like *ENOD40A*, *NIN2A*, and *ERN1b* (upregulated specifically during nodulation[49–52]) in *Rj2 rfg1* Sil$_{G3Pdh}$ roots inoculated with U122 within 24 h (Supplementary Fig. 10b).

The compatibility phenotype of *Rj2 rfg1* Sil$_{G3Pdh}$ plants with U122 was associated with their inability to accumulate leaf G3P in response to U122 infection (Fig. 4g). As expected, the Sil$_{G3Pdh}$ plants were also defective in *Psg avrB*-induced leaf G3P accumulation (Fig. 4g). Consistent with the importance of SA and G3P in SAR, the Sil$_{ICS}$, Sil$_{PAL}$ and Sil$_{G3Pdh}$ plants were defective in the onset of both pathogen- and incompatible rhizobia-induced SAR (Fig. 4h, i). There was no significant change in the number or size of nodules produced by compatible rhizobia on Sil$_{G3Pdh}$ plants in the *Rj2*, *Rfg1* or *rj2 rfg1* backgrounds (Fig. 4b, Supplementary Fig. 11). Together, these data showed that G3Pdh-derived G3P is essential for *Rj2/Rfg1*-mediated strain-specific exclusion of incompatible rhizobia, whereas both SA and G3P were required for *Psg avrB*- or U122-induced systemic immunity. However, the specific rhizobium infection stage at which G3P enables *Rj2/Rfg1*-incompatibility is unclear.

**Rhizobia incompatibility induces G3P accumulation in vascular exudate**. We next asked if systemic signaling in response to incompatible rhizobia was associated with G3P accumulation in

the vascular tissue. We assayed G3P levels in the exudate collected from leaf (Pex) or roots (Rex) of U122-inoculated plants. Inoculation with U122-induced G3P in both Pex and Rex collected from *Rj2* plants (Fig. 5a). In contrast, Pex and Rex did not show any increase in either AzA or SA (Supplementary Fig. 12). Application of Rex from U122-treated *Rj2* plants (Rex$_{U122}$) on the leaves of a fresh set of soybean plants induced systemic resistance against *Psg Vir* (*Psg Vir* was infiltrated into leaves distal to those infiltrated with Rex$_{U122}$), whereas Rex from U257-treated *Rj2* plants (Rex$_{U257}$) did not (Fig. 5b). The Rex$_{U122}$-induced systemic resistance was as robust as that induced by Pex from *Psg avrB*-infected plants (Pex$_{avrB}$). Importantly, neither heat nor proteinase K treatments diminished the systemic resistance inducing activity of Rex$_{U122}$ (Fig. 5c), suggesting the involvement of non-proteinaceous components in this response. As expected, Sil$_{G3Pdh}$ plants were defective in both pathogen- and incompatible rhizobia-induced G3P accumulation in Pex/Rex (Fig. 5d). However, Rex from Sil$_{G3Pdh}$ plants was able to confer systemic resistance in a fresh set of wild-type soybean plants (Rex$_{U122}$ from Sil$_{G3Pdh}$ plants was infiltrated into leaves of wild-type plants followed by *Psg Vir* infection in distal leaves) (Fig. 5e). Together, these results suggested that the root induced signal associated with the induction of foliar resistance was unlikely to be G3P.

**Foliar G3P is essential for rhizobia incompatibility**. Root inoculation with incompatible rhizobia-induced systemic resistance, which was associated with foliar accumulation of G3P. Therefore, we tested if systemic G3P accumulation was important for rhizobium incompatibility using grafting assays. Grafting between *Rj2* and *rj2* scions and rootstocks showed that incompatibility was determined by the genotype of the rootstock; U122 only produced nodules on *rj2* rootstocks, regardless of the scion genotype (*Rj2* or *rj2*). Conversely, U122 did not produce any nodules on *Rj2* rootstocks regardless if they were grafted to *Rj2* or *rj2* scions (Fig. 6a). This was also true for U122-induced systemic resistance. Inoculation with U122 on *Rj2* rootstocks grafted to either *Rj2* or *rj2* scions induced foliar resistance to *Psg*, but not when U122 was inoculated on *rj2* rootstocks grafted to *Rj2* or *rj2* scions (Fig. 6b). Thus, root genotype was important for excluding incompatible rhizobia as well as induction of foliar resistance.

We then generated grafts between *Rj2* V and *Rj2* Sil$_{G3Pdh}$ rootstocks and scions followed by root inoculation with U122. qPCR analysis showed appropriate gene-silencing specificity; *GmG3Pdh* was knocked down specifically only in the Sil$_{G3Pdh}$, but not V, rootstocks or scions (Supplementary Fig. 13). As expected, U122 produced nodules on *Rj2* Sil$_{G3Pdh}$ rootstocks grafted to *Rj2*

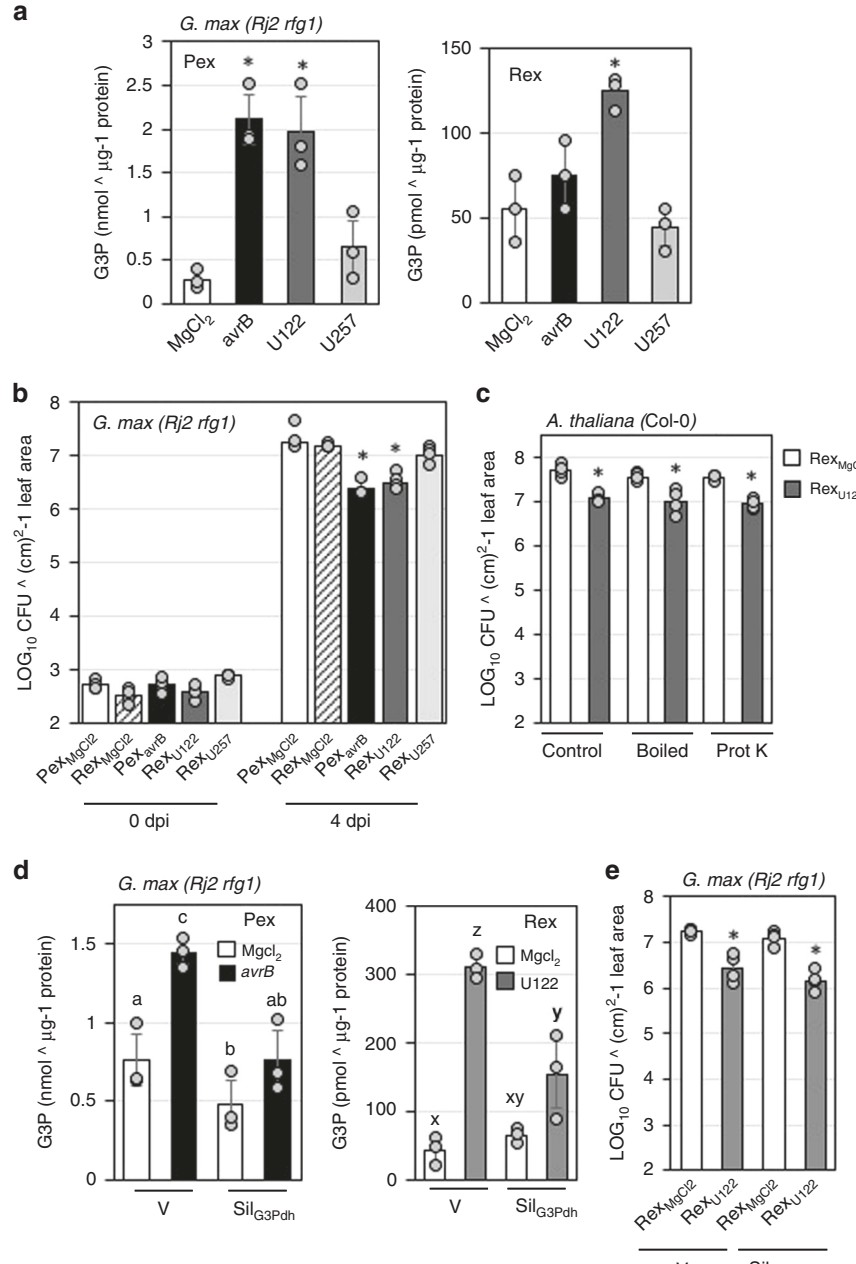

**Fig. 5** Glycerol-3-phosphate accumulates in the vascular exudates of soybean plants in response to incompatible rhizobia. **a** G3P levels in petiole (Pex) or root (Rex) exudate of *Rj2 rfg1* plants 24 h post inoculation with buffer (MgCl2), compatible (U257), incompatible (U122) rhizobia, or *Psg avrB* (*avrB*). Asterisks denote significant differences from MgCl2 of corresponding genotype, Student's *t*-test, $P < 0.0001$ ($n = 4$). **b, c** *Psg Vir* counts presented as LOG10 values of colony forming units (CFU) ^ cm2-1 leaf area from infected leaves at 0 and 4 days post inoculation (dpi). Error bars indicate standard deviation ($n = 4$). Asterisks denote data significantly different from corresponding MgCl2, Student's *t*-test, $P < 0.001$. **b** Pex or Rex from buffer (Pex/RexMgCl2), U122 (RexU122), or *Psg avrB* (PexavrB) inoculated *Rj2 rfg1* plants was leaf-infiltrated in a fresh set of *Rj2 rfg1* plants followed by *Psg Vir* inoculation on systemic leaves 48 h later. **c** Untreated (control), boiled, or proteinase K-treated (Prot K) Rex MgCl2/U122 was leaf-infiltrated in *A. thaliana* plants (Col-0) followed by *Pst Vir* inoculation on systemic leaves. **d** G3P levels in Pex or Rex of V and SilG3Pdh *Rj2 rfg1* plants 24 h post inoculation with MgCl2, *avrB*, or U122. Letters denote significant differences based on two-way ANOVA in SAS ($P < 0.05$ after Tukey correction). **e** *Psg Vir* counts presented as LOG10 values of (CFU) ^ cm2-1 leaf area from infected leaves at 0 and 4 dpi. Error bars indicate standard deviation ($n = 4$). Asterisks denote data significantly different from corresponding mock, Student's *t*-test, $P < 0.001$. RexMgCl2/U122 from V (plants infected with control VIGS vector) or SilG3Pdh (*GmG3Pdh* knockdown) *Rj2 rfg1* plants was leaf-infiltrated in wild-type *Rj2 rfg1* or *rj2 rfg1* plants followed by *Psg Vir* inoculation on systemic leaves. Results are representative of two-four independent experiments.

SilG3Pdh scions but not on *Rj2* V rootstocks grafted to *Rj2* V scions (Fig. 6c). This indicated that knockdown of *G3Pdh* expression was able to inhibit *Rj2*-mediated exclusion of U122 in graft tissue. Interestingly, U122 produced nodules on *Rj2* V rootstocks grafted to *Rj2* SilG3Pdh scions, but not when *Rj2*

SilG3Pdh rootstocks were grafted to *Rj2* V scions (Fig. 6c). Together these data suggested that G3P production in the leaves was important for strain-specific exclusion of incompatible rhizobia, although recognition of incompatibility occurred in the root.

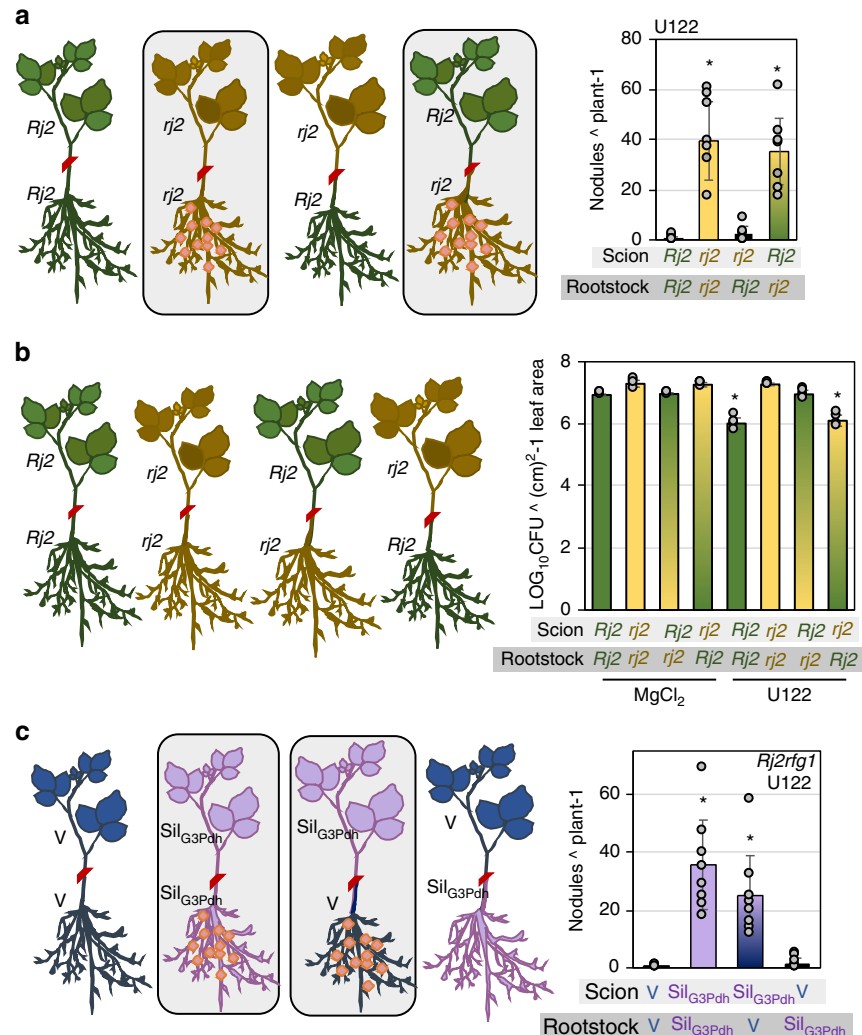

**Fig. 6** Leaf-derived G3P is required for root exclusion of incompatible rhizobia. **a** Average number of nodules produced by U122 on grafts between *Rj2* and *rj2* genotypes. Genotype of rootstocks and scions are indicated in gray boxes below the *x*-axis. Boxed panels indicate grafts that produced nodules in response to U122. **b** *Psg Vir* counts on *Rj2* and *rj2* grafts pretreated (root) with buffer (MgCl$_2$) or U122. *Psg Vir* was inoculated on scions 48 h after pretreatment of rootstocks with U122. LOG$_{10}$ values of (CFU) colony forming units (cm) 2$^{-1}$ leaf area from infected leaves at 0 and 4 days post inoculation (dpi) are presented. Error bars indicate standard deviation ($n = 4$). Asterisks denote significant difference between mock and U122 plants of corresponding genotypes, Student's *t*-test, $P < 0.005$. **c** Average number of nodules produced by U122 on grafts between control VIGS vector inoculated (V) and G3Pdh knockdown (Sil$_{G3Pdh}$) *Rj2* plants. Asterisks denote significant differences from V × V grafts, *t*-test, $P < 0.001$.

**Incompatible rhizobia induce shoot to root transport of G3P.** In order for leaf-derived G3P to contribute to the exclusion of incompatible rhizobia in the root, G3P produced in the leaves must be transported to the root. We tested this using exogenous G3P application assays. Leaf infiltration of G3P neither affected the incompatibility of U122, nor compatibility of U257 on *Rj2* plants (Fig. 7a). Likewise, G3P did not alter the compatibility of U122 on *rj2* plants (Fig. 7a, right panel). However, G3P infiltration did inhibit U122 nodulation on *Rj2* Sil$_{G3Pdh}$ plants; U122 produced significantly fewer nodules on G3P-infiltrated versus MgCl$_2$-infiltrated *Rj2* Sil$_{G3Pdh}$ (Fig. 7b). These results supported the notion that leaf G3P was transported to root only in response to incompatible rhizobia, where it inhibited root nodulation. To test this further, we assayed the accumulation of $^{14}$C in the roots of rhizobia-inoculated plants that were leaf-infiltrated with $^{14}$C-G3P. The extent of G3P transport was measured as the amount of $^{14}$C detected in stem or root extracts 48 h post inoculation with incompatible rhizobia. Interestingly, significantly more $^{14}$C was detected in both stem and roots of plants inoculated with

incompatible than with compatible rhizobia (Fig. 7c). Whole plant imaging studies also showed $^{14}$C in the stem and roots of incompatible rhizobia-, but not compatible rhizobia-inoculated plants, that were leaf-infiltrated with $^{14}$C-G3P (Fig. 7d). Thin-layer chromatography (TLC) of their root extracts showed that G3P was transported as one or more derivatives (Fig. 7e) and that some of it was converted to glycerol (black arrowhead) likely due to G3P phosphatase activity[30]. Notably, however, pathogen infection resulted in negligible root transport of G3P as compared with incompatible rhizobia and consequently foliar pathogen (*Psg avrB*) infection was unable to inhibit root nodulation by compatible rhizobia (Supplementary Fig. 14). Importantly, the TLC separated G3P and derivatives from root extracts were able to induce systemic resistance when infiltrated into Arabidopsis plants (Fig. 7f).

**Discussion**

Based on the data presented, we hypothesize that inoculation of incompatible rhizobia (do not make nodules) results in the

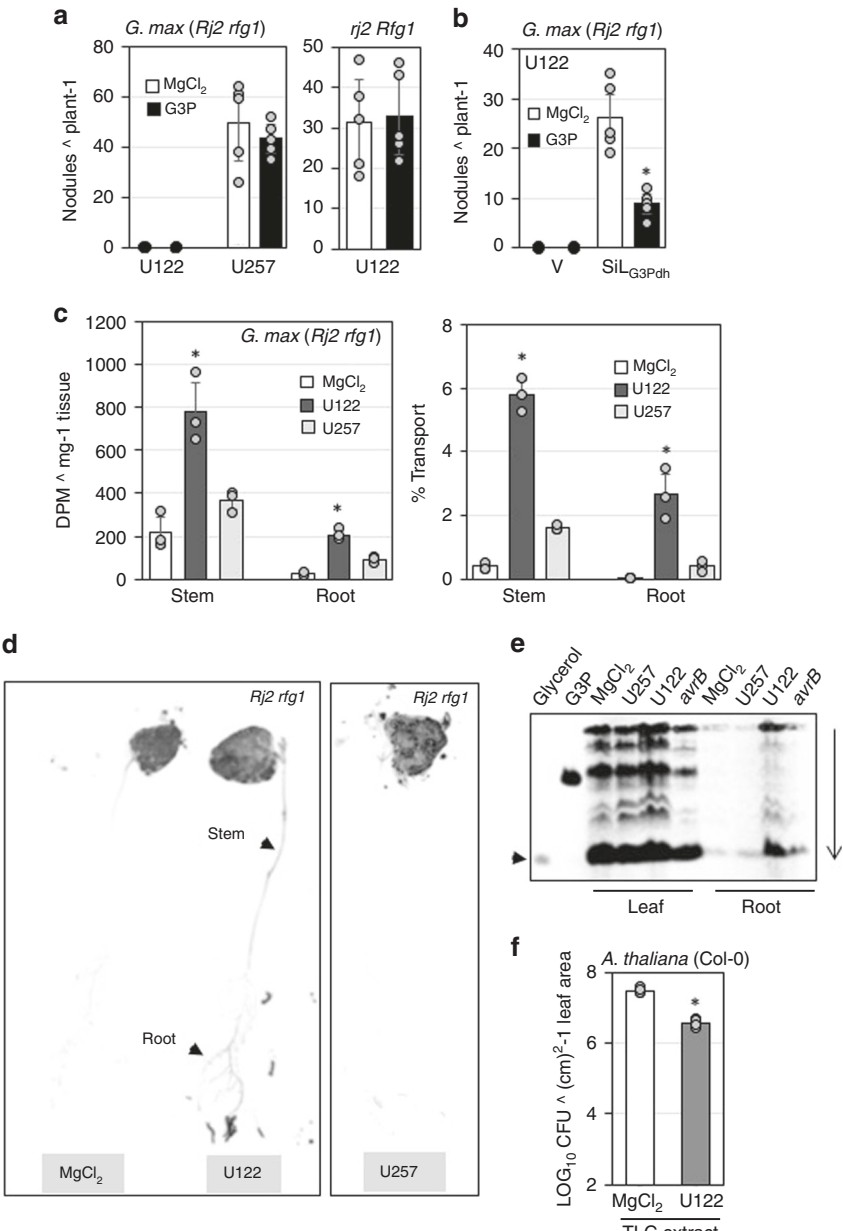

**Fig. 7** Incompatible rhizobia induce shoot to root transport of G3P. **a**, **b** Average number of nodules produced by U122 or U257 on **a** wild-type, **b** control VIGS vector inoculated (V) or G3Pdh knockdown (Sil$_{G3Pdh}$) *Rj2 rfg1* plants, that were leaf-infiltrated with MgCl$_2$ or G3P 24 h after rhizobia inoculation. Black circles on x axis indicates absence of nodules. **c** Quantification of radioactivity (left panel) and percentage $^{14}$C-G3P transported (right panel) to stem and roots of *Rj2 rfg1* plants that were root-inoculated with buffer (MgCl$_2$), U122 (incompatible), or U257 (compatible) followed by leaf infiltration with $^{14}$C-G3P. $^{14}$C levels were measured 48 h later. **d** Autoradiograph of *Rj2 rfg1* seedlings that were root-inoculated with MgCl$_2$, U122 or U257 followed by $^{14}$C-G3P infiltration in leaves. Images were obtained 24 h post $^{14}$C-G3P infiltration. **e** Autoradiograph of thin-layer chromatogram (TLC) of leaf and root extracts prepared from plants that were root-inoculated with MgCl$_2$, U122 or U257 followed by leaf infiltration with $^{14}$C-G3P. $^{14}$C-G3P and $^{14}$C-glycerol were loaded as controls. The extracts were prepared 48 h post rhizobia inoculation and chromatographed on a cellulose plate. Vertical arrow indicates direction of run. Arrowhead indicates glycerol. **f** *Pst Vir* counts in leaves of *Arabidopsis thaliana* (Col-0) plants that were pre-infiltrated with TLC separated root extracts from MgCl$_2$ or U122-inoculated *Rj2 rfg1* soybean plants as in (**e**). Root extracts from *Rj2 rfg1* plants inoculated with MgCl$_2$ or U122 (root) were separated on TLC. TLC extracts were infiltrated into *A. thaliana* plants followed by *Pst Vir* inoculation on systemic leaves, 48 h later. LOG$_{10}$ values of colony forming units (CFU) ˆ cm$^2$-1 leaf area are presented. Error bars indicate standard deviation (n = 4). Asterisks denote data significantly different from corresponding mock, Student's t-test, P < 0.005. Results are representative of two-four independent experiments.

generation of a non-proteinaceous, heat tolerant signal other than G3P, which is transported from the root to the shoot, where it induces G3P biosynthesis. G3P translocates back to the roots as G3P and one or more derivatives and restricts nodulation by undesirable rhizobia. The underlying pathway overlaps with SAR signaling because it involves the accumulation of ROS, AzA, G3P,

and SA in the foliar tissue and knockdown of G3P biosynthesis compromises both incompatible rhizobia and pathogen-induced systemic resistance. Thus, the incompatible rhizobia-mediated systemic signaling is distinct from induced systemic resistance, which is typically activated in response to infection by select compatible strains of plant growth-promoting organisms like

*Pseudomonas*, *Bacillus*, *Trichoderma*, and mycorrhiza species[53]. We propose that a unique root-shoot-root signaling pathway, mediated in part by G3P, provides protection against foliar pathogens while restricting root colonization by less-desirable symbionts.

## Methods

**Plant growth conditions**. Soybean (*Glycine max* (L.) Merr.) cvs L76-1988 (*Rj2 rfg1*), L82-257 (*rj2 Rfg1*) Harosoy (*Rpg1-b*), and Peking (*rj2 rfg1*) were grown in the green house with day and night temperatures of 25 and 20 °C, respectively. L76-1988 also has the *Rpg1-b* genotype based on pathogenicity assays with *Psg avrB*. Plants were grown in sterile vermiculite or in steamed soil and roots of plants (V1 stage[54]) were inoculated with *B. japonicum* USDA122 or *S. fredii* USDA257 (0.2 OD$_{600}$). Nodulation phenotypes were recorded 2 weeks post inoculation by uprooting plants and counting nodules. Microscopic analyses of nodule development was done using light microscopy (×40) or confocal microscopy of roots treated with the cell-permeant green florescent nucleic acid-binding dye STYO®13[55]. Briefly, U122-infected soybean roots were harvested at 2 or 3 dpi (root hair curling) and 6 dpi (nodule primordia) and washed with 50 mM PIPES buffer (pH 7.0). Roots were then stained with STYO®13 (1 μl in 10 ml 50 mM PIPES buffer) for 15 mins. ×40 images were obtained using laser scanning confocal microscopy. Root hair curling and nodule primordia counts were done using light microscopy (×40) of secondary roots ~10 cm away from the root-shoot junction. Root hair curling measurements were made for at least 3 cm per root with approximately six plants per genotype. Nodule primordia measurements were made for 2–6 cm per root for approximately six plants per genotype.

Nodulation-specific gene expression was recorded by harvesting root tissue for RNA extraction. For silencing experiments, soybean seedlings at VC stage were inoculated with recombinant Bean pod mottle virus (BPMV) vectors and confirmation of silencing was carried out using qPCR[38–41]. Plants inoculated with a VIGS vector containing a non-specific sequence (V[38]) were used as control for all VIGS-related experiments.

**Pathogen strains and inoculations**. *Pseudomonas syringae* pv. *glycinea* race four expressing AvrB[56], via the broad host range plasmid pDSK519 was used. The strain expressing the empty pDSK519 plasmid was used as *Vir* control. *Psg* strains were grown on King's B medium at 28 °C, supplemented with rifampicin 50 mg ml$^{-1}$ plus kanamycin 50 mg ml$^{-1}$. *Psg* inoculation of soybean was done using pressure infiltration[47] and bacterial proliferation was monitored at 0 and 4 dpi. Mock inoculations were carried out with 10 mM MgCl$_2$ in 0.04% Silwett L-77. Results are representative of three to four independent repeats, unless noted otherwise. For pathogen-induced SAR in soybean, leaves of V1 plants were infiltrated with either MgCl$_2$ or *Psg avrB* (10$^7$ CFU ml$^{-1}$). Forty eight hours later, the systemic leaves were infiltrated with *Psg Vir* (10$^5$ CFU ml$^{-1}$). Growth of *Psg Vir* was monitored at 0 and 4 dpi[46]. For rhizobia-induced systemic resistance, soybean roots were inoculated with 0.5 OD of compatible or incompatible rhizobia. Forty eight hours later, leaves were infiltrated with *Psg Vir* (10$^5$ CFU ml$^{-1}$) and growth was monitored at 0 and 4 days post inoculation (dpi). For SAR analysis in Arabidopsis, 4-week old Arabidopsis plants (Col-0) were leaf-infiltrated with buffer (MgCl$_2$), *Pst avrRpt2* (10$^7$ CFU ml$^{-1}$), or petiole exudate from leaves of *Psg avrB*-infected/ MgCl$_2$-infiltated soybean plants. Forty eight hours later, the systemic leaves were infiltrated with *Pst* DC3000 (10$^5$ CFU ml$^{-1}$). Growth of *Pst* DC3000 was monitored at 0 and 3 dpi[30].

**Generating gene knockdown vector and knockdown plants**. *GmG3Pdh* gene sequence (225 bp fragment encoding the W83-E157 region of GmG3Pdh, primer sequences in Supplementary Table 1) was cloned into pGG7R2V using sequence-specific primers linked to *Bam*HI (forward primer) and *Msc*I (reverse primer) sites and ligated to the *Bam*HI/*Msc*I digested pGG7R2V. pGG7R2V is the cloned RNA2 of Bean Pod Mottle Virus[38]. The recombinant RNA2 was in vitro transcribed and inoculated along with in vitro transcribed RNA1 of a mild (Hancock) BPMV strain. Once infectious virus had established on the plant and silencing of target gene was confirmed, infected tissue was freeze-dried and used as inoculum for subsequent inoculations. For each new inoculation silencing was confirmed before performing the specific experiment. Virus/transcripts were inoculated on the first true leaves at VC stage.

**RNA extraction and quantitative RT-PCR**. RNA from leaf/root tissues of soybean plants at V2/V3 growth stage was extracted using the TRIzol reagent (Invitrogen, Carlsbad, CA), per manufacturer's instructions. Reverse transcription (RT) and first strand cDNA synthesis were carried out using Superscript II (Invitrogen, Carlsbad, CA). Two to three independent RNA preparations were analyzed by quantitative RT-PCR to evaluate relative differences in transcript levels. Primers were designed to amplify gene specific PCR products of <200 bp in size. Actin was employed as an internal control to normalize the cDNA. qRT-PCR was carried out in 96-well plate using SYBR Green Mix[47]. Cycle threshold values were calculated by SDS 2.3 software. Gene expression was quantified using the relative quantification (ΔCt) method[57]. Each sample or treatment was tested in at least three biological

repeats and the same experiment was performed twice. Primers used for qPCR analysis are listed in primer sequences in Supplementary Table 1.

**Grafting in soybean**. Grafting was done in plants at V1 stage. Cotyledonary node and cotyledonary leaf (unifoliate leaf) was removed from the rootstock. Then, rootstalk above cotyledons axis was split vertically and grafted with a wedge-shaped scion bearing the developing leaf shoot. Grafts were secured in place by wrapping with parafilm. Grafted plants were covered with paper bags with holes and sprayed with water at 2–3 h intervals for a week. After week-long acclimatization, paper bags were replaced by transparent plastic bags, which were removed gradually over the course of the second week. Grafts were root-inoculated with rhizobia or leaf inoculated with *Psg Vir*. Gene silencing was assessed by quantitative RT-PCR analysis of RNA from scion (leaf) and rootstock (root) tissue.

**G3P quantifications, and mobility assays**. G3P[58] was extracted from ~1 g leaf tissues. Leaf tissue was frozen in liquid nitrogen and homogenized in 5 mL 80% (v/v) ethanol containing 100 μM 2 deoxy-glucose as an internal standard. The extracts were analyzed on ICS3000 high-performance liquid chromatography (HPLC) fitted with PA1 column (Dionex Inc., IL).

For in planta G3P mobility assays, soybean seedlings were root-inoculated with 25 mL of 0.5 OD$_{600}$ U122 or U257. Twenty four hours later, unifoliate leaves were infiltrated with 80 μM of $^{14}$C-G3P. For *Psg avrB*-induced G3P mobility assays, *Psg avrB* (10$^7$ CFU ml$^{-1}$) was co-infiltrated with $^{14}$C-G3P into leaves followed by sampling of leaf, stem and root tissue sampled 24 h later. For $^{14}$C quantification, tissue was weighed and $^{14}$C extractions were done in 300 μl of water. Amount of $^{14}$C in the extract was quantified using a liquid scintillation analyzer. For thin-layer chromatography (TLC), samples were run on pre-coated cellulose plates (0.1 mm; EM Laboratories) using n-butanol:acetic acid:water (2:1:1 vol) and autoradiographed using Typhoon PhosphorImager. For whole plant imaging, leaves were infiltrated with $^{14}$C-G3P or $^{14}$C-G3P + *Psg avrB* (10$^7$ CFU ml$^{-1}$), and whole seedlings were autoradiographed 24 h post leaf infiltration. For bioactivity of TLC separated G3P derivatives, root extract from U122-inoculated *Rj2* plants infiltrated with cold G3P was run parallel to extract from $^{14}$C-G3P-infiltrated plants. Cold G3P TLC lane corresponding to the $^{14}$C-G3P run was eluted using n-butanol:acetic acid:water (2:1:1 by vol), dried under a stream of nitrogen gas and resuspended in ~500 μl ml$^{-1}$ of deionized water. This was infiltrated in leaves of Arabidopsis plants followed by infiltration of *Pst* DC3000 (10$^5$ CFU ml$^{-1}$) on systemic leaves 48 h later. Growth of *Pst* DC3000 was monitored at 0 and 3 dpi.

**Preparation of and treatments with petiole/root exudate**. Petiole exudates[58] were collected from soybean plants (*Rj2*) induced for SAR either by root inoculation with U122/U257 (0.5 OD$_{600}$) or by foliar infiltration of *Psg avrB* (10$^7$ CFU ml$^{-1}$). Twenty four hours later, petioles and roots were excised, surface sterilized in 50% ethanol and 0.0006% bleach, rinsed in sterile 1 mM EDTA and submerged in 20 ml of 1 mM EDTA and 100 μg/ml ampicillin. Petiole and root exudates were collected over 48 h and infiltrated into healthy Arabidopsis plants (Col-0) for SAR measurement. For proteinase K treatments, petiole exudate from U122-inoculated *Rj2* plants was incubated with 60 μg/ml proteinase K for 2 h at 37 °C. For heat treatment, petiole exudate from U122-inoculated *Rj2* plants was incubated for 30 min in a boiling water bath.

**SA, ROS, AzA, quantification**. SA was extracted and measured from ~100 mg of fresh weight (FW) tissue[58]. Plant tissue was homogenized in 90% methanol, the homogenate centrifuged and the pellet was reextracted using 100% methanol. The pooled supernatant was dried under N$_2$ gas, resuspended in 5% trichloroacetic acid (2.5 mL), incubated on a shaker for 30 min. Free and conjugated SA were separated in the organic and aqueous phases, respectively, via organic extraction with two volumes of ethylacetate-cyclopentane-isopropanol (50:50:1) and separated using high-performance liquid chromatography.

AzA was extracted and measured from ~150 mg FW tissue[58]. Briefly, tissue was homogenized in 1 ml of chloroform:methanol (2:1 v/v), containing 5 μg sebasic acid as an internal standard. Two hundred microliters of glacial acetic acid and 1 ml of 0.9% KCl were added to the homogenate, vortexed vigorously for 5 s and centrifuged for 1 min. The lower phase was collected in a new glass tube and reextracted with 1 ml of chloroform. The extracts were methylated with 4.8% sodium methoxide and acidified with glacial acetic acid. After adding 1 ml of 0.9% KCl samples were reextracted with chloroform, evaporated under a stream of nitrogen gas, methylated with diazomethane, and finally dried under a stream of nitrogen gas. The dried samples were suspended in 200 μL of iso-octane, transferred to gas chromatography (GC) vials, and analyzed on a GC-mass spectrometer (MS).

ROS were measured using electron spin resonance spectroscopy (ESR) using ~150 mg fresh weight tissue[58]. Briefly, leaves were homogenized in HEPES buffer (pH 6.9) containing 50 mM 4-POBN [α-(4-pyridyl-1-oxide)-N-tert-butylnitrone]. Ten microliters of the homogenate was loaded onto graduated capillary tube in a flat cell. ESR spectra were measured at room temperature using a Bruker ESP 300 X-band spectrometer set at 5 mW microwave power, 100 kHz modulation frequency, 1G modulation amplitude, and 9.687 GHz microware frequency. Values of ESR signals were calculated from the maximum-signal/noise ratio of recorder

traces and corrected, if necessary, by subtracting reagent blanks determined in parallel. Signal intensity was evaluated as the peak height in ESR spectra.

**Sequence accessions and analysis.** Database accessions for sequences used here are GmG3Pdh (Glyma02g38310), PR1 (Glyma13g251600), NR (Glyma14g165000), DFR (Glyma17g173200). Sequence alignment and phylogenetic analysis were carried out using the Megalign program in the DNASTAR package[59].

**Yeast complementation assay.** For yeast complementation assay[60] full-length soybean G3Pdh gene was inserted into the EcoRI-PstI sites of pEG202 vector and transformed into the S. cerevisiae gpd1 mutant (host strain BY4741). Wild-type yeast and gpd1 mutant transformed with empty pEG202 were used as positive and negative controls, respectively. Transformants (wild type + pEG202, gpd1 + pEG202, and gpd1 + pEG202-GmG3Pdh) were selected on synthetic minimal medium containing all essential amino acids except histidine (selection marker of pEG202). Transformants and untransformed wild-type yeast were grown in yeast potato dextrose (YPD) medium until cell density reached $OD_{600} = 1.0$. 10 μl of the serially diluted (1:10 dilutions in the range of $10^4–10^6$ CFU ml$^{-1}$) cultures were spotted onto YPD medium containing 1 M NaCl. Growth was monitored for 3–4 days at 30 °C.

**Transcriptome analysis.** The total RNA integrity or quality was checked (Agilent 2100 Bioanalyser, Agilent, USA) and raw RNA-Seq data was obtained through 100 cycle HISeq high-output mode sequence method per lane. Sequence quality was assessed using FastQC (http://www.bioinformatics.babraham.ac.uk/projects/fastqc/, v0.10.1) for all samples. The paired end reads were then mapped against the STAR indexed reference genome[61] of Glycine max (a2.v1) downloaded from Phytozome 11[62]. Splice aware mapping algorithm STAR[63] (v2.5.2a) was used for map RNA-seq reads to the reference genome using default parameters. For assigning sequence reads to the genomics features, feature counts from the Subread[64] (v1.4.6) package was used. Only primary alignments, ignoring multi-mapped and chimeric reads were used for generating counts. Counts from all the samples were converted to a desirable format using AWK command and differential gene expression analyses (DGE) was carried out by EdgeR[65] (v3.14.0) using negative binomial and generalized linear models. The DGE is expressed as Log2FC and are considered significant if the false discovery rate (FDR) was <0.05. Information for the differentially expressed genes were paired from the official gene annotations obtained from Phytozome to help identify relevant trends. A total of 3 such comparisons were performed comparing control with USDA122, control with USDA257, or USDA122 with USDA257. The authors acknowledge the support of the Genome Informatics Facility, Office of Biotechnology at Iowa State University, for RNA-seq analysis.

**Reporting summary.** Further information on research design is available in the Nature Research Reporting Summary linked to this article.

## Data availability
All data presented here and biological materials used are available upon request to the corresponding author. The source data underlying Figs. 1–7 and Supplementary Figs. 2–14 are provided as a Source Data file. Microscopy data is available via Figshare at https://doi.org/10.6084/m9.figshare.9976829. RNA-seq data are available at the NCBI GEO database under accession code GSE139303.

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

## Acknowledgements

We thank Dr. Hongyang Zhu (L76-1988/L82-257 soybean seeds and USDA122/257 strains), Dr. Carl Bradley (amplifying soybean seed stock), Dr. Emily Pfeuffer (statistical analyses), Dr. Anne-Frances Miller (ESR analysis) and Amy Crume (management of plant growth facilities) for their generous help. This work was supported by NSF (IOS #1457121), USDA National Institute of Food and Agriculture (Hatch project 1014539), and Kentucky Soybean Promotion Board grants to A.K. and P.K. and faculty start-up funds by Iowa State University to A.K.S.

## Author contributions

M.B.S. and Q.-.m.G. conducted experiments and contributed equally. R.V.G.-.R. conducted RNA-seq analysis. A.K.S. oversaw RNA-seq analysis. P.K. conducted metabolite analysis. M.B.S. and A.K. analyzed and prepared data. A.K. conceived the project. A.K., P.K., and M.B.S. prepared the paper. A.K., P.K., and A.K.S. edited the paper.

## Competing interests

The authors declare no competing interests.
