## [Peer Review File · Nature Communications]

Reviewers' comments:

Reviewer #1 (Remarks to the Author):

This manuscript aims to investigate the possible role of systemic acquired resistance induced by rhizobia known to be incompatible with soybean carrying the Rj2 TIR-NB-LRR R-gene. The involvement of glycerol-3-phosphate known to provide systemic immunity in foliar infection is suggested to serve as shoot to root mobile signal in rhizobial incompatibility.

The questions raised are interesting and there are results suggesting that a novel contribution can eventually be developed from the approach taken. However, as stands the investigation presented in the manuscript is too preliminary and has experimental and methodological issues. The investigation and presentation is also suffering from shallow integration of the results into our current mechanistic understanding of immunity and compatibility, see comments below.

Major comments.

Introduction:

The description of the plant genetics and recognition mechanisms governing compatibility and incompatibility as presented in the introduction is superficial and not up-to-date. Given the many investigations addressing immunity/compatibility in soybean and other legumes the focus on early day plant lectin suggestions seems outdated, see Sugawara et al 2018, Okazki et al PNAS 2013, Nature Comms, Yang et al PNAS 2010, papers from the Hongyan Zhu group, Zypfel and Oldroyd 2017 for review and papers from the Ratet P and Kondorosi groups. There is now a large body of evidence on the molecular function of the Rj2, Rj4 and Rfg1 genes in incompatibility that is largely ignored and incomplete referenced, see above. Another example, results showing the role of bacterial surface polysaccharides (Kawaharada et al Nature 2015; Kawaharada et al Ncomms 2017) are ignored. This part of the introduction should be rewritten to set the scene with current knowledge on key components and the relevant molecular mechanisms.

Phenotyping: The phenotyping of rhizobial interaction and nodule development presented throughout this manuscript fall short of the standard in the field. Compatibility between a host and a rhizobial bacteria under different experimental conditions/treatments should be characterized and determined by quantifying subsequent steps in nodule development from root hair curling, formation of infection pockets, development of infection threads, formation of nodule primordia and finally fully developed root nodules. Only the latter is presented for the different VIGS knockdowns and this is insufficient to draw firm conclusions on the level of compatibility and mechanistic implications resulting from the VIGS knockdown. This is important since the manuscript concludes that "recognition of incompatibility occurs in the root", the question is at what stage of the bacterial signalling/infection?

Most experiments in this manuscripts lacks a proper control only U122 on Rj2 results are presented. Inoculation with U257 is occasionally used. However, a line with recessive rj2 allele which is the obvious control that makes a link directly to the R-gene mediated SAR is missing. Fig 1 also needs to include results from plants that were not treated with the empty VIGS.

Mechanisms: The observed compatibility changes following RAR1 and NDR1 knockdown is interesting but is not followed up by detailed mechanistic investigations of consequence for immunity and this disconnects this set of results from the rest of the manuscript and the RAR1 and NDR1 mediated molecular mechanism involved in immunity. A similar disconnect to current mechanism regulating early Nod factor mediated signalling and autoregulation of nodule numbers

by root to shoot signalling appears on page 8. Here, Nod- and a Nod+ bacterial strains are tested on SG3Pdh plants while the more relevant connection to known components of the root-shoot-root signaling of the autoregulatory system is not investigated.

Methodology:

The metabolic analyses needs to be described, the method used is not obvious to the reader. Page 6 and Fig 2, how were the level of G3P, AzA and SA measured? Fig 2 h lacks detailed explanation in Materials and methods.

The transport assay presented in Fig 6 is not convincing. There are no controls like wild type plants or even better rj2 plants included and this is crucial.

Minor comments:

Legend to Suppl Fig 2 suggest that results from knockdown in both Rj2 and Rfg1 plants are shown but only one set of results are shown.

Fig 7 not needed.

Reviewer #2 (Remarks to the Author):

Review:

Summary:

The authors provide evidence that incompatible rhizobia that are associated with soybean roots induce defense responses (in a Rj2, Rfg1 dependent manner) that require signaling components (i.e RAR1 and NDR1) of known disease resistance pathways that protect against pathogens. Activation of these pathways also results in the induction of systemic acquired resistance (SAR) and accumulation of SAR-associated metabolites. A key finding is that glycerol-3-phosphate (G3P) is required for restriction of rhizobia infection in roots during incompatible interactions and for restriction of pathogenic bacteria in leaves. They also provide evidence that G3P produced in the shoot is required to restrict the incompatible rhizobia and suggest that shoot to root transport of G3P is involved.

Critique:

This an important paper. It is trying to bring two fields together – symbiosis signaling and disease resistance signaling. Linking a role for the SAR metabolite, G3P, in restricting rhizobia invasion in roots and pathogenic bacteria in shoots demonstrates that while host recognition genes are specific, some of the key the down-stream signaling components are shared and operate exclude both types of bacteria.

The impact of this work would be strengthened with additional controls/experiments to support the key finds and editorial changes/corrections to better communicate the findings for a diverse audience. Key considerations:

Experimental:

1. The authors provide evidence that silencing G3Pdh reduces protection by lowering G3P levels and application of petiole extracts from induced soybean plants (incompatible reaction) provides protection. They have not shown that application of G3P alone is sufficient to provide protection and prevent nodulation formation. The authors used similar approaches to verify G3P action in Arabidopsis (Chanda et al 2008).

In addition, GFP treatment of the grafted plants in Figure 5 should be done for chemical complementation. Application to the shoots vs roots will help support shoot to root trafficking of G3P.

2. Figure 1c needs to be better integrated with 1a and 1b. In the text they are separated. The key point here is that incompatible interactions, elicited by pseudomonas and rhizobia, restrict bacterial growth and this is dependent on Rj2. The authors need to clearly label the genotype of the plants (Rj2, rfg1) and include the appropriate genetic control (rj2, rfg1).

3. Figure 2a is missing a key control. Need to isolate Pex from soybean plants that are not infected with avrB.

4. Figure 3 should include the root phenotypes of the other treatments.

Editorial:

5. The title is a bit misleading. G3P is required, but it has not been shown to be sufficient. Regulation is a strong word without supporting data. Would be helpful to indicate this is soybean or legumes.

6. Plant and Genotype labeling throughout the manuscript is mixed. Clear labels on all figures should be present so that you don't have to read the legend for key info. Info presented below will try to clue in confusion areas.

7. Figure 1 confusion:

(a) Give full genotype for host plants: Rj2, rfg1; rj2, Rfg1. Indicate number tested directly: 26 nodules, n=8.

(b) Inserts of nodules are not clear and need to explicitly show "insert".

(c) Spell out mock and don't use "M". Clearly indicate plant genotype.

8. Figure 2 confusion:

(a) Label this as Arabidopsis. Confusing use of Pex: P = petiole, and then later use P = shoot.

(b-d) Label as soybean.

(e-g) Label as soybean leaves.

(h) Get rid of abstract spectra. Not clear, especially without axis labels and a control mock.

9. Figure 3 confusion:

(a) This is not a good way to present findings. Move to supplement or perhaps consider not including as it is difficult to parse out findings. More importantly, it is labelled as Psg. You need to clarify "Psg" vs "Psg avrB" and clearly indicate which data is yours and which is the published data.

(b) Should just tell the reader that you looked for G3Pdh based on your G3P work. Obvious. Then the figure makes sense. Again label all subfigures with plant genotype. Also, SG3Pdh label is confusing. I suggest: Sil-G3Pdh for silencing.

10. Figure 4 confusion:

(d) Stats is not done properly. This should be a 2-way Anova if you are comparing black bars. Star vs letter is not standard.

11. Figure 5 confusion:

Hard figure to digest. Keep in mind: S=Susceptibility or Shoot... R=Resistance or Root. Confusing for readers. Consider using a different label or use a legend.

12. Figure 7 is not useful. Remove.

13. Supplemental figure 1 should be better sighted. It also is missing significant research in the Pipecolic acid pathway. It is misleading to the readers, as much more is known and it is not presented or cited.

14. Also, in many places in the work it is not known where the samples are coming from: leaves, or roots, or petioles of mixture.

We thank both reviewers for their critical review of our manuscript and the helpful suggestions. Following is a detailed response to reviewers comments.

Reviewers' comments:

Reviewer #1 (Remarks to the Author):

This manuscript aims to investigate the possible role of systemic acquired resistance induced by rhizobia known to be incompatible with soybean carrying the Rj2 TIR-NB-LRR R-gene. The involvement of glycerol-3-phosphate known to provide systemic immunity in foliar infection is suggested to serve as shoot to root mobile signal in rhizobial incompatibility.

The questions raised are interesting and there are results suggesting that a novel contribution can eventually be developed from the approach taken. However, as stands the investigation presented in the manuscript is too preliminary and has experimental and methodological issues. The investigation and presentation is also suffering from shallow integration of the results into our current mechanistic understanding of immunity and compatibility, see comments below.

We thank the reviewer for their critical analysis of our manuscript and the helpful suggestions for improvement. We have incorporated all the suggested textual changes and appropriate experimental data.

Major comments.

Introduction:

The description of the plant genetics and recognition mechanisms governing compatibility and incompatibility as presented in the introduction is superficial and not up-to-date. Given the many investigations addressing immunity/compatibility in soybean and other legumes the focus on early day plant lectin suggestions seems outdated, see Sugawara et al 2018, Okazki et al PNAS 2013, Nature Comms, Yang et al PNAS 2010, papers from the Hongyan Zhu group, Zypfel and Oldroyd 2017 for review and papers from the Ratet P and Kondorosi groups. There is now a large body of evidence on the molecular function of the Rj2, Rj4 and Rfg1 genes in incompatibility that is largely ignored and incomplete referenced, see above. Another example, results showing the role of bacterial surface polysaccharides (Kawaharada et al Nature 2015; Kawaharada et al Ncomms 2017) are ignored. This part of the introduction should be rewritten to set the scene with current knowledge on key components and the relevant molecular mechanisms.

We apologize for the oversight in excluding relevant publications and have rewritten the introduction as suggested.

Phenotyping: The phenotyping of rhizobial interaction and nodule development presented

throughout this manuscript fall short of the standard in the field. Compatibility between a host and a rhizobial bacteria under different experimental conditions/treatments should be characterized and determined by quantifying subsequent steps in nodule development from root hair curling, formation of infection pockets, development of infection treads, formation of nodule primordia and finally fully developed root nodules. Only the latter is presented for the different VIGS knockdowns and this is insufficient to draw firm conclusions on the level of compatibility and mechanistic implications resulting from the VIGS knockdown. This is important since the manuscript concludes that “recognition of incompatibility occurs in the root”, the question is at what stage of the bacterial signalling/infection?

As suggested we have included early phenotypes associated with nodulation.

Most experiments in this manuscripts lacks a proper control only U122 on Rj2 results are presented. Inoculation with U257 is occasionally used. However, a line with recessive *rj2* allele which is the obvious control that makes a link directly to the R-gene mediated SAR is missing. Fig 1 also needs to include results from plants that were not treated with the empty VIGS. We included compatible controls for all experiments as suggested. For Fig 1c we added the *rj2* control as suggested. For all other analyses, we used a compatible strain (U257) on *Rj2* because it allows us to compare various effects in the same genetic background. This is especially important for metabolite analyses where even basal levels can vary significantly amongst different cultivars.

Fig 1: Data for the mock (no VIGS vector) control is added in the revised submission. Images for compatible bacteria on the VIGS control (U122 on *rj2 Rfg1 V*) are included. Data for *avrB*/U122/U257-induced SAR in *rj2 Rfg1* and *rj2 rfg1* plants is also included in the revised manuscript.

Fig 2: Includes data for compatible (U257) and incompatible (U122) interactions in *Rj2*

Fig 3: Includes compatibility data (U257 on *Rj2*, or U122 on *rj2*) for the relevant panels.

Fig 4: Compatibility data (U257 on *Rj2*) added for 4b in the revised submission.

Fig 5: Includes *rj2* controls.

Fig 6: Compatibility data (U257 on *Rj2*) added for 6c-6e in the revised submission.

Mechanisms: The observed compatibility changes following RAR1 and NDR1 knockdown is interesting but is not followed up by detailed mechanistic investigations of consequence for immunity and this disconnects this set of results from the rest of the manuscript and the RAR1 and NDR1 mediated molecular mechanism involved in immunity. A similar disconnect to current mechanism regulating early Nod factor mediated signalling and autoregulation of nodule numbers by root to shoot signalling appears on page 8. Here, Nod- and a Nod+ bacterial strains are tested on SG3Pdh plants while the more relevant connection to known components of the root-shoot-root signaling of the autoregulatory system is not investigated.

Based on reviewer's comments we included the foliar resistance phenotypes for RAR1 and NDR1 silenced lines (revised Fig 1d, 1e). Our data are in agreement with the demonstrated requirement of these components in pathogen-induced SAR in Arabidopsis. Please note that the roles of RAR1 and NDR1 in R mediated signaling are well documented. For instance, RAR1 is well known to regulate R protein stability.

Both Nod and AON are relevant to compatible host-rhizobia interactions, whereas our current work focuses on genetically driven incompatible host-rhizobia interactions. Please note that G3P has no effect on compatible interactions as tested here. Therefore, we have removed the data for Nod and Nod receptor mutants from this manuscript. We do agree that it would be important to test the involvement of G3P in the AON pathway and thank the reviewer for the suggestion. We have requested AON-related mutant lines from Peter Gresshoff's laboratory and he has kindly agreed to share. However, we hope the reviewer will appreciate that in-depth analyses of all relevant mutants will require substantial more time and effort.

Methodology:

The metabolic analyses needs to be described, the method used is not obvious to the reader. Page 6 and Fig 2, how were the level of G3P, AzA and SA measured? Fig 2 h lacks detailed explanation in Materials and methods.

We apologize for the oversight, better methodology details are included for all metabolite analyses in the revised manuscript.

The transport assay presented in Fig 6 is not convincing. There are no controls like wild type plants or even better *rj2* plants included and this is crucial.

All transport assays were in fact done in wild type plants. We have included data for U257 which is compatible on *Rj2* (revised Fig 6a, 6c-6e). We believe this is a better control for compatibility than U122 on *rj2* because it allows us to test G3P transport in response to compatible and incompatible rhizobia in the same genetic background.

Minor comments:

Legend to Suppl Fig 2 suggest that results from knockdown in both *Rj2* and *Rfg1* plants are shown but only one set of results are shown.

The figure legend was corrected as suggested.

Fig 7 not needed.

Figure 7 was removed as suggested.

Reviewer #2 (Remarks to the Author):

Review:

Summary:

The authors provide evidence that incompatible rhizobia that are associated with soybean roots induce defense responses (in a *Rj2*, *Rfg1* dependent manner) that require signaling components (i.e *RAR1* and *NDR1*) of known disease resistance pathways that protect against pathogens. Activation of these pathways also results in the induction of systemic acquired resistance (SAR)

and accumulation of SAR-associated metabolites. A key finding is that glycerol-3-phosphate (G3P) is required for restriction of rhizobia infection in roots during incompatible interactions and for restriction of pathogenic bacteria in leaves. They also provide evidence that G3P produced in the shoot is required to restrict the incompatible rhizobia and suggest that shoot to root transport of G3P is involved.

Critique:

This an important paper. It is trying to bring two fields together – symbiosis signaling and disease resistance signaling. Linking a role for the SAR metabolite, G3P, in restricting rhizobia invasion in roots and pathogenic bacteria in shoots demonstrates that while host recognition genes are specific, some of the key the down-stream signaling components are shared and operate exclude both types of bacteria.

The impact of this work would be strengthened with additional controls/experiments to support the key finds and editorial changes/corrections to better communicate the findings for a diverse audience. Key considerations:

We thank the reviewer for their critical analysis of our manuscript and the helpful suggestions for improvement. We have incorporated all suggested changes.

Experimental:

1. The authors provide evidence that silencing G3Pdh reduces protection by lowering G3P levels and application of petiole extracts from induced soybean plants (incompatible reaction) provides protection. They have not shown that application of G3P alone is sufficient to provide protection and prevent nodulation formation. The authors used similar approaches to verify G3P action in *Arabidopsis* (Chanda et al 2008).

In addition, GFP treatment of the grafted plants in Figure 5 should be done for chemical complementation. Application to the shoots vs roots will help support shoot to root trafficking of G3P.

Based on reviewer's comments we included data for exogenous G3P application as suggested (revised Figs 6a, 6b). Note that G3P application alone is not sufficient to inhibit nodulation by compatible bacteria. However, localized G3P infiltration into leaves did inhibit root nodulation by incompatible rhizobia in *Sil_{G3Pdh}* plants. These results support our hypothesis that recognition of incompatibility through Rj2 is important to induce the overall signaling cascade which enables transport of leaf G3P to the root.

We also tried localized G3P infiltration in leaves of *Sil_{G3Pdh}* graft tissue. However, there are technical difficulties with this experiment as virus-infected leaves on graft scions are unable to withstand multiple pressure infiltrations (unlike *Arabidopsis*, we must use pressure infiltration on the trichome dense soybean leaves). The graft tissue collapsed soon after pressure infiltration of G3P or $MgCl_2$.

2. Figure 1c needs to be better integrated with 1a and 1b. In the text they are separated. The key point here is that incompatible interactions, elicited by *Pseudomonas* and rhizobia, restrict

bacterial growth and this is dependent on Rj2. The authors need to clearly label the genotype of the plants (Rj2, rfg1) and include the appropriate genetic control (rj2, rfg1).

The text and subsections were modified to better integrate Fig 1c with rest of the figure as suggested. All plant genotypes were labelled as suggested. The *rj2 rfg1* control was included as suggested.

3. Figure 2a is missing a key control. Need to isolate Pex from soybean plants that are not infected with avrB.

This control was included in our original experiments though not presented in the original submission. We apologize for the oversight. These data are now presented in the revised manuscript.

4. Figure 3 should include the root phenotypes of the other treatments.

We presume the reviewer means nodulation phenotypes for Sil_{ICS} and Sil_{PAL}. These images are included in the revised manuscript. Images for Mock and V plants are in Figure 1.

Editorial:

5. The title is a bit misleading. G3P is required, but it has not been shown to be sufficient. Regulation is a strong word without supporting data. Would be helpful to indicate this is soybean or legumes.

Based on reviewer's comments the title was changed to ***Glycerol-3-phosphate mediates novel rhizobia-induced systemic signaling in soybean***

6. Plant and Genotype labeling throughout the manuscript is mixed. Clear labels on all figures should be present so that you don't have to read the legend for key info. Info presented below will try to clue in confusion areas.

As suggested, we clarified the plant species and genotype in all figures.

7. Figure 1 confusion:

(a) Give full genotype for host plants: Rj2, rfg1; rj2, Rfg1. Indicate number tested directly: 26 nodules, n=8. Done as suggested. The subscript number indicated standard deviation. It is now presented as average number nodules (+/- standard deviation), n=10-15 plants as indicated in figure legend.

(b) Inserts of nodules are not clear and need to explicitly show "insert". Done as suggested

(c) Spell out mock and don't use "M". Clearly indicate plant genotype. Done as suggested

8. Figure 2 confusion:

(a) Label this as Arabidopsis. Confusing use of Pex: P = petiole, and then later use P = shoot. Done as suggested. Pex only indicates petiole exudate. To avoid confusion with shoot, we changed "shoot" to "leaves/leaf" wherever appropriate in the text.

(b-d) Label as soybean. Done as suggested

(e-g) Label as soybean leaves. Done as suggested

(h) Get rid of abstract spectra. Not clear, especially without axis labels and a control mock. Done as suggested

9. Figure 3 confusion:

(a) This is not a good way to present findings. Move to supplement or perhaps consider not including as it is difficult to parse out findings. More importantly, it is labelled as Psg. You need to clarify “Psg” vs “Psg avrB” and clearly indicate which data is yours and which is the published data.

(b) Should just tell the reader that you looked for G3Pdh based on your G3P work. Obvious. Then the figure makes sense. Again label all subfigures with plant genotype. Also, SG3Pdh label is confusing. I suggest: Sil-G3Pdh for silencing.

Based on reviewer’s comments we removed data comparing transcriptional changes between incompatible rhizobia (our data) and pathogen-infected (published work) plants. We also moved part of the original Fig 3a to Supplemental Fig 4a, changed S_{G3Pdh} to Sil_{G3Pdh} and labeled all subfigures with plant genotypes for Fig 3 as well as all other figures.

10. Figure 4 confusion:

(d) Stats is not done properly. This should be a 2-way Anova if you are comparing black bars. Star vs letter is not standard.

Two-way Anova was used in SAS, as suggested.

11. Figure 5 confusion:

Hard figure to digest. Keep in mind: S=Susceptibility or Shoot... R=Resistance or Root. Confusing for readers. Consider using a different label or use a legend.

Based on reviewer’s comments the R/S labeling was changed to scion and rootstock.

12. Figure 7 is not useful. Remove.

Figure 7 was removed as suggested.

13. Supplemental figure 1 should be better sighted. It also is missing significant research in the Pipecolic acid pathway. It is misleading to the readers, as much more is known and it is not presented or cited.

As suggested, Supplemental Figure 1 was modified to include pipecolic acid, pipecolic acid-related SAR work (including some of our recently published work) was cited, and this figure was better referenced in the text.

14. Also, in many places in the work it is not known where the samples are coming from: leaves, or roots, or petioles of mixture.

The source of tissue samples for all experiments was clarified as suggested.

Reviewers' comments:

Reviewer #1 (Remarks to the Author):

The changes and new data included in the revision has improved the manuscript but there are still important issues that should be addressed.

The revised introduction now present an acceptable introduction to our current understanding of legume-rhizobia compatibility. In order to provide the reader a better perspective on shoot derived signals it would be informative to include the original Science and Nature Comms references for the role of miR2111 and cytokinin as shoot-derived signals rather than a review¹², see page 3. On the same page the sentence "The dominant Rj/Rfg genes do not resemble the "loss of function" recessive Nod-factor receptors" does not make sense and should be replaced. A simple statement that the Nod-factor receptors are LysM-RLK belonging to a different receptor family than Rj/Rfg will do and again provide references to the original Nature papers.

The phenotyping has been improved but is still falling short of the current standard in the field. Images of root hairs have now been added in Figure 1a, however the quality is poor and better images should be included. The rj2Rfg1/U122 image show two root hairs curling around each other rather than a rhizobia induced curling. To understand at which infection or developmental stage the Rj2-U122 incompatibility is established and overcome by the SILNDR1/RAR1 it is necessary to quantify the phenotypic observations. For example: How many infection pockets, infection threads and nodule primordia per cm root in controls and treatments. The same comment goes for Fig 3 d, quantitative phenotypic data is required to place incompatibility control in a developmental context and relate the observations to our understanding of infection and nodule organogenic processes.

We thank the reviewer for their comments, following is a detailed response to reviewer concerns.

Reviewers' comments:

Reviewer #1 (Remarks to the Author):

The changes and new data included in the revision has improved the manuscript but there are still important issues that should be addressed.

The revised introduction now present an acceptable introduction to our current understanding of legume-rhizobia compatibility. In order to provide the reader a better perspective on shoot derived signals it would be informative to include the original Science and Nature Comms references for the role of miR2111 and cytokinin as shoot-derived signals rather than a review¹², see page 3. On the same page the sentence “The dominant Rj/Rfg genes do not resemble the “loss of function” recessive Nod-factor receptors” does not make sense and should be replaced. A simple statement that the Nod-factor receptors are LysM-RLK belonging to a different receptor family than Rj/Rfg will do and again provide references to the original Nature papers.

-The text related to Rj/Rfg genes was corrected as suggested. Original articles for miR2111/cytokinin as shoot-derived signals and Nod-factor receptors as LysM-RLK type genes were included in the revised manuscript (Reference nos 12, 13, 22, 23).

The phenotyping has been improved but is still falling short of the current standard in the field. Images of root hairs have now been added in Figure 1a, however the quality is poor and better images should be included. The rj2Rfg1/U122 image show two root hairs curling around each other rather than a rhizobia induced curling. To understand at which infection or developmental stage the Rj2-U122 incompatibility is established and overcome by the SILNDR1/RAR1 it is necessary to quantify the phenotypic observations. For example: How many infection pockets, infection threads and nodule primordia per cm root in controls and treatments. The same comment goes for Fig 3 d, quantitative phenotypic data is required to place incompatibility control in a developmental context and relate the observations to our understanding of infection and nodule organogenic processes.

-Based on reviewer comments we included graphs (Figure 1c) or table (Supplemental Figure 9a) showing quantitative data for infection pockets and nodule primordia per cm root in control and RAR1/NDR1/G3Pdh knockdown plants. In addition, (based on Editor suggestions) we also included additional images for root hair curling, nodule density and nodule morphology (Supplemental Figures 3, 4 & 9) to better illustrate the observed phenotypes.